# To Send a Kite: Simone Weil's Lessons in Ethical Attention for the Curator

**Maggie Sava** 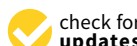

Independent Researcher, Denver, CO 80203, USA; mmsava@gmail.com

**Abstract:** As socially engaged practices grow within the curatorial field, the use of attention becomes a crucial ethical decision. How and to whom attention is given centers on concerns of visibility, belonging, and the determination of those characteristics within a community's negotiated communicative space. Exploring Simone Weil's ethics of attention through and alongside incarceration-focused curatorial projects, this article positions her writing as a potential framework for attentive curation. The resulting pathways found in Weil's writing offer means of transforming the curatorial into a self-silencing act of witnessing that serves underrecognized voices. This research parses how Weilian attention redefines inquiry as the process of listening to and incorporating others' perspectives as primary sources of knowledge. Looking towards an ethics of Weilian attention with examples of incarceration-focused curation reveals how upholding the insights and articulations of marginalized individuals promotes social wellbeing and works towards the realization of justice. Thousand Kites, a prison-based project connecting inmates and the public through the radio and internet, provides the central case study for a curatorial project aligning with Weilian attention.

**Keywords:** Simone Weil; attention; ethics; curatorial ethics; curation; incarceration; prisons

---

## 1. Introduction

The compounding political, economic, and social forces in the United States prison-industrial complex propagate a racialized carceral system that, as Angela Davis states in her seminal text *Are Prisons Obsolete?* stands to make "huge profits from processes of social destruction" [1] (p. 88). The rise in prison-privatization and the presence of corporations in prisons have caused a portion of the economy to rely on the constant supply of inmates as sources of cheap labor and profit [1] (p. 84). A disproportionate number of people experiencing incarceration are Black, Indigenous, and Latino/a/x individuals who serve sentences inequivalent to their wealthy, white counterparts [2].[1] The structures of the prison-industrial complex benefit from inmates not having the same rights protecting them as non-incarcerated citizens while denying them many of the means of participation in society at large. Mass incarceration in the United States destabilizes communities and families. This disruption of social support can increase the likelihood of those affected becoming involved in the criminal justice system [1] (pp. 16–17). In its most basic function, incarceration isolates people from the public and makes them virtually invisible. However, images and representations of prisons and prison life exist as a major component of American visual culture and mass media to the point where they have become a ubiquitous and normalized reality [1] (pp. 15–19). Davis presents the contradiction of invisibility and visibility relating to the prison–industrial complex: "We take prisons for granted but are often afraid to

---

[1]   A Pew Research study from 2010 found that "black men were more than six times as likely" and "Hispanic men were nearly three times as likely" as white men to be incarcerated in the United States. The disparity was similar within the female population [2].

face the realities they produce" [1] (p. 15). Within this reflection, Davis underscores questions regarding who enacts exposure through representation, how it shapes the ways people witness the realities of suffering, and how those factors may actually serve the interests of existing oppressive forces.

Educators, community organizers, and even prison officials have for decades recognized the opportunities and the need to employ art as a response to the not-always-transparent realities of incarceration that have become entrenched in the American cultural landscape. Prison-based arts have come to make up a substantial portion of participatory and community arts initiatives. As a 2014 report for the National Endowment of the Arts found, however, decreased government funding beginning in the mid-1990s and increased prison privatization, which disincentivizes the maintenance of such initiatives due to the cost, has reduced the amount of art programming within prisons [3] (p. 7). In spite of this, art practitioners and advocates have continued to bring attention to the benefits of arts in prisons. Through partnerships with curators, museums, and galleries, they have hosted several exhibitions to bring greater visibility to inmates' work and the conditions they face while imprisoned.

Interrogations of witnessing and exposure, while not new, remain integral to prison-based curatorial practices. They continuously navigate social attention and critically examine its effect on narrations of experience as a way to expose the realities of the oppressive and appropriative structures that silence marginalized incarcerated voices. Individuals in the prison system have complicated stories that do not create a uniform and universalizing schema. Authenticity is vital when addressing the injustices that infiltrate and obscure the recognition of inmates' personhood. Many curators working in and around prisons seek ways to break from existing paradigms of exposition in order to foster genuine and generative forms of witnessing. Discussions about how to curate such issues of suffering, silencing, and need for recognition against suppressive forces are ongoing within curatorial theory. While this paper does not offer a singular resolution, it intends to contribute to the conversation by offering up an ethical framework built upon Simone Weil's unorthodox philosophy of attention. Her writing can help inspire new forms of witnessing by changing how the curatorial is understood concerning accounts of precarity and vulnerability. Weil asserts the obligations of being for others and attending to the affliction of those who experience the greatest material deprivation, political adversity, and prejudice without diluting their realities. The unwavering confrontation of hardship at the center of Weil's writing connects the abstract, immaterial needs of recognition and communicative agency with lived community-based participation. Her thought contributes novel views of how cultural work can create potential sites for resistance and stewardship.

> Writing in a style that is simultaneously philosophical and prayerful, Weil desires:
>
> May I disappear in order that those things that I see may become perfect in their beauty from the very fact that they are no longer things that I see.
>
> I do not in the least wish that this created world should fade from my view, but I do wish that it should no longer be shown to me in person. To me it cannot tell its secret which is too high. If I go, then the creator and the creature will exchange their secrets [4].
>
> (p. 89)[2]

Weil's humbling appeal of transparency provides a framework for attention as "thought that is pure," or prayer-like thought [5] (p. 192), which curators can adopt to promote another's ability to express and share their reality as it informs different experiences of the world. The process of selecting and presenting can carve out spaces and designate resources for encounters akin to sharing

---

[2]　This excerpt of Weil's writing is taken from *Gravity and Grace*, a collection of her notebooks written before May of 1942 and compiled and edited by her friend Gustave Thibon. At that time, Europe, including her home country of France following its defeat by Nazi forces, was afflicted by state-level violence against which Weil had previously written and spoken out while working with factory laborers, advocating for the working class, and temporarily joining the anarchist forces in the Spanish Civil War. *Gravity and Grace* presents an outlook and attempt of being in the world in a way that departed from the desolation and harm that she witnessed emerging from political, social, and economic wielding of power [4] (pp. vii–43).

"secrets" where participants can engage in unadulterated communicative acts not just with curators but with audiences and the public as well. The resulting approach involves relinquishing oneself to another in the understanding that the moment of contact is not for the curator's sake but rather for the non-assimilable participant.

This paper develops a definition of attentive curating as a social justice-oriented practice focused on centering voices dampened or silenced by systems of subjection or marginalization [6].[3] It is not a response to, but a witnessing and an upholding of the other's call to be recognized. Attentive curation holds a delicate and challenging relationship with exposure: it is built around and enhances it, and yet it also attempts to mediate the risk, the harm, and the vulnerability that exposure can create for the other. It involves an ability to detach from a curator's personal interests, the nature and criteria of the spaces in which they work (including goals of profit and marketability), and the expectations of the artists or participants preceding the encounter so that those no longer serve as overarching objectives. It reflects on how those culminating factors potentially recreate and enact harmful power dynamics and thus removes oneself from them so that the curatorial event serves the other's needs. It is a way of seeing curating not as a perfectible technique or a repeatable pattern. Instead, it is always destabilized or overturned by the acknowledgment of the other. It is an inherently flexible format best suited for practitioners whose work can occur independently and without strict preconditions.

The following discussion begins by addressing how Weilian attention reorients how one accesses and learns about the world. It requires seeing others as the origin and access point of unique, essential perspectives of reality. Accordingly, it preserves the integrity of their voices to prevent appropriation or alteration. The paper then turns back to the ethical demand made of the curator and their responsibilities as the silent listener—namely, the need to relinquish the self. As it builds a portrait of attentive curation, this paper looks at how incarceration-focused curation involves complex intersections of suffering, silencing, and calls to witness. Examining methods used in past projects illuminates how attentive practices can enrich ethical encounters between curators and prisoners. In order to identify ethics echoing Weilian attention at play in curation, the essay offers the central case study of Thousand Kites, initiated by Nick Szuberla and Amelia Kirby and carried out through Appalshop. The storytelling initiative shares the accounts and creative works of inmates living in Supermax facilities in Appalachia. By organizing available resources and finding the most accessible channels through which to project incarcerated voices, Szuberla and Kirby's efforts demonstrate how communication and meaning-making can provide an attentive movement for justice.

## 2. To Serve as a Witness

Weil's petition for disappearance, which permits "that those things that I see may become perfect in their beauty from the very fact that they are no longer things that I see," reveals a foundational desire to give over space to that which exists beyond and around her through abdication of self. From her particular position, the world she sees cannot achieve full realization or expression, as "To me it cannot tell its secret which is too high." Her perspective and presence create a limitation, and true revelation only becomes possible if the obstacle of her subjective frame is removed. However, Weil maintains a tension in wanting to disappear and not wanting to be fully absent, best expressed in her statement that she does not "in the least wish that this created world should fade from my view, but I do wish that it should no longer be shown to me in person." In her near self-contradiction, Weil communicates a wish that refuses the polarity of an "I" affirming presence versus complete non-existence, a wish to exist in

---

[3]　　A discussion of the large, existing body of curatorial work and theory addressing questions of social justice and equality exceeds the limits of this paper. The focus on incarceration-based curation in the United States creates a defined field of curatorial investigations as a parameter for the development of attentive curating. For a wide-ranging discussion of contemporary practices and exhibitions handling issues of racism, sexism, and other outstanding forms of inequality, Maura Reilly's *Curatorial Activism: Towards an Ethics of Curating* provides an examination of curatorial ethics and an extensive bibliography with further resources [6].

an alternative manner as a blank conduit serving the profound exchange she describes [7] (Chapter 4, About her departure, para. 12).[4] The hope to witness in such an intimate manner, made possible by self-restraint, silence, and pause, underlines Weil's notion of attention.

Weil posits that attention, achieved through discipline and habit formation [5] (p. 206), makes an individual fundamentally receptive. Although she views prayer to God as its highest form, she recognizes its role in learning about and interacting with the world in its social, political, and material manifestations [8] (p. 199), [9] (p. 95).[5] Essentially, it informs how an ethical actor seeks awareness. Attention is not exclusively about an end goal of obtaining knowledge. Rather, it radically restructures how one accesses it in a way that, as Peter Roberts describes, becomes "both an epistemological and a moral process" [10] (p. 321). Unadulterated attention prepares one to receive the central essence or message of either the idea, entity, or the person on which one focuses. The attentive individual can then act, as Weil articulates, as a medium for the "radiant manifestation of reality" [11] (p. 242).

Weil addresses the relational responsibilities of attention by establishing a distance between the self as the individual practicing attention and the other as to whom attention is directed. She emphasizes this unbreachable space not as a means of creating a hierarchy or justifying oppression, silencing, or violence based on difference. Instead, Weil stresses that the other cannot be known through the self. Attentive practices acknowledge their distinctive, innate presence. She states that "to soil is to modify, it is to touch. The beautiful is that which we cannot wish to change. To assume power over is to soil. To possess is to soil" [4] (pp. 114–115). Attention operating from a distance allows for a non-assimilating and non-invasive act of witnessing because, as Weil describes, it refuses to decide others' readings. While "readings," "reading," and "being read" are familiar and broadly used terms in contemporary scholarship, they bear special meaning in Weil's work. Gustave Thibon defines her notion of reading as "emotional interpretation, the concrete judgment of value" [4] (p. 188). Weil problematizes evaluative reading as it can create a limitation of the other. Her discussion of attention instead offers a means of heeding the other without intruding on or containing their presence by focusing on the impersonal. The impersonal does not involve personality traits or characteristics of the other, which tend to be the aspects of a person that are most prevalently read [12].[6] It embodies the desire for good and hope for justice expressed in a person's foundational and inviolable essence [13] (p. 70). Justice is, in Weil's view, "to be ever ready to admit that another person is something quite different from what we read when he is there (or when we think about him)" [4] (p. 188). If "every being cries out silently to be read differently," justice means allowing others continuous opportunities to be seen in new ways [4] (p. 188). In turn, attention resists interpretive control and singularity.

The way Weilian attention redefines intellectual and social inquiry has significant implications for curatorial projects handling issues of suffering and vulnerability. The restrained nature of Weilian attention and the circumstances of witnessing it poses differ from the productive and transactional attention encouraged in traditional work and educational environments. It emphasizes non-expectancy and makes the individual ever ready to receive the call made by someone who needs their precarity to be recognized without it being altered or diluted. Reading is a familiar concept for curating, which requires similar work through selecting and arranging materials, objects, and stories. It can shift emotional, social, cultural, intellectual, and even monetary "value." However, by decentering the curator's voice, attention changes the communicative roles so that the primary influence in meaning-making moves

---

[4] As Lyndsey Stonebridge states, experiencing the world through this silent presence is "to discover another way of being in it" [7] (Chapter 4, About her departure, para. 12).

[5] Stuart Jesson clarifies the expanded implications of Weilian attention by explaining that "a love of the world is a love of God" [8] (p. 199). Offering a similar understanding, Mary G. Dietz points out that Weilian attention can help gain "a deeper insight on things and events in the world" [9] (p. 95).

[6] As Cynthia Gayman explains, the impersonal is not the "interrelational self" [12] (p. 196), but that which exists beyond the specific and the social [12] (pp. 200, 202), "not an attribute of the self, but an implicit orientation of the self towards others and, ultimately, toward the good" [12] (p. 197).

from the curator to the participants. Through open receptivity, the role and duties of the curatorial position patiently wait to be created and shaped by the other's articulation and their call for justice.

Movement in the direction of a similar curatorial approach emerges in the exhibition *Lift Us Up, Don't Push Us Out* (*Lift Us Up*), held at the Art League Gallery in Alexandria, Virginia from 9 January to 3 February 2019 [14,15].[7] Organized by Performing Statistics, Art 180, and the Legal Aid Justice Center, and curated by Mark Strandquist, *Lift Us Up* connects youth from the Richmond Juvenile Detention Center with artists and lawyers as they create artworks about their experiences of incarceration. The focus of the exhibition, located in a state with high youth incarceration rates, is the school-to-prison pipeline crisis. In the pipeline, disciplinary measures, which disproportionately affect Black, Indigenous, and Latino/a/x youth, LGBTQIA+ youth, and youth with disabilities, siphon children from classrooms into detention centers starting at young ages [15,16]. Annie Green, writing for the Art League, states that the primary drive of the show is to demonstrate "the importance of listening to and learning from incarcerated members of society to see them as experts in the conversation surrounding the prison system" because they are "the most important and relevant voices in answering the question: 'How can we keep kids free?'" [15]. One of the strongest ways that the project takes on the call to listen to and uphold the perspectives of those incarcerated without speaking for them is by including police manuals that the participants create. Local officers attend the show and directly face the manual's request to recognize the youth as important community members before seeing them as criminals so that they may be read in new and generative ways. Such recognition would allow them to be supported rather than, as Green puts, "held back by circumstances often beyond their control" [15]. The youth participants also compose a curriculum about the school-to-prison pipeline, which goes out to schools for teachers to incorporate into their lessons [17]. They offer their own stories as forms of knowledge and suggest steps for reform based on their own requirements for justice while the curator and organizers serve as what Strandquist identifies as "human megaphones for these teens whose voices have been systemically silenced" [17]. Through *Lift Us Up*, children living in the Richmond Juvenile Detention Center become primary experts and educators about youth incarceration while the curatorial component acts as a channel for the expression of their experiences and their desire to be recognized.

## 3. Attentive Justice and the Social Implications of Self-Silencing

Ceding communicative primacy works towards justice in expanded readings by silencing the self. An attentive individual can reach a state of such openness through what Weil identifies as decreation. The decreative ethical demand that Weil's writing forms comes from the exposure of the other's vulnerability. In this relationship, the other's call for recognition conjures the attentive listener who does not have a self-driven purpose outside of this call, just as they do not see Weil's proverbial landscape for themselves [18] (p. 36). Although seemingly writing against Weil's ethical call, Emmanuel Levinas offers a closer discernment of decreation's implications by questioning:

> Does divine goodness consist in treating man with an infinite pity that lies within this supernatural compassion that moves Simone Weil, or in admitting him into His Society, and treating him with respect? To love one's neighbour can mean already to glimpse his misery and rottenness, but it can also mean to see his face, his mastery over us, and the dignity he has as someone who is associated with God and has rights over us [19].

(p. 139)

---

7    The 2019 exhibition held at the Art League Gallery is only one of the project's several iterations. The 2018 show, first hosted at ART 180's Atlas Gallery, was organized to travel to different venues, including schools, museums, police departments, and public spaces [14].

Levinas argues that, rather than suffering being the source of virtue, "The just man who suffers is worthy not because of his suffering but because of his justice, which defies suffering" [19] (p. 141). By emphasizing that seeing the other entails a revealing of that which exists beyond their affliction and has "mastery over us," and that the exposure is what creates the obligation to the other, Levinas speaks to the ethical bind at play in Weil's work that demands the surrendering of the self to the other's inviolable call for justice. The difference in Levinas's take is that he suggests that the responsibility to the other occurs alternatively to or in spite of suffering while suffering in Weil's framework is the universal condition that facilitates attention and obligation to the other.

Weil asserts that the only thing that the attentive individual can truly give to the other is their "I," or their subjectivity, which they surrender through decreation [4] (pp. 119, 165). It is "to die in order to liberate a *tied up* energy, in order to possess an energy which is free and capable of understanding the true relationship of things," which enhances attention and witnessing [4] (p. 81), [20] (p. 213).[8] Decreation remains one of Weil's most challenging and abstract concepts, though, because it seems implausible. It not only aims to rid one of their fundamental "I,"—it also appears to repudiate the value of personal attributes and particularity. However, decreation becomes practicable when understood as a denial of the presumed authoritative position of the "I" rather than complete self-annihilation. It creates an awareness of one's own potential precarity and susceptibility to the uncontrollable force which inheres suffering [21] (p. 29).[9] The result is an obligation to attend to another's suffering that can stifle their communicative power and obscure recognition of their personhood. Decreation cultivates such witnessing by restraining the power that the attentive self has to speak over or to diminish the other's voice and delivering it over to the other.

Unanswered, however, as Yoon Sook Cha carefully uncovers in *Decreation and the Ethical Bind*, is the question as to whether or not that gift of the "I," or the ego, can be fully accomplished or, if so, whether that self-surrender can sufficiently serve the other's call [18] (p. 63). The ethical obligation remains incomplete and perpetual, as the others' vulnerability can never be entirely resolved. This does not always engender a negative sense of failure, as the incompleteness also continuously refuses domain over the other and prevents a potential collapse of the distance between the self and the other resulting from the notion of a completed favor. Importantly, understanding the incompleteness of the call would not equate to an acceptance of the inevitability of the other's suffering and passivity that allows it to continue unaddressed. Again, Levinas's response to Weil's work offers a way to understand the ethical obligation. He stresses the terms act and action multiple times and contends that "interiority," maintains an incomplete abstraction of God's universality [19] (pp. 136–137). As he states, "We do not conceive of relations, we are in relation. It is not a question of inner mediation, but of action. There is no redemption of the world, only a transformation of the world" [19] (pp. 140–141). A decreative relationship, in asking for the gift of the "I," puts the "I" into the service of the other who calls for attention and protection, even if it may not be completed. It remains a self-silencing transformation while still permitting action because it does not act on behalf of the attentive individual nor at their direction—it uses its ability to interrupt the force harming the other so that they may have the agency to proliferate alternative readings of themselves and their experiences. Seeing someone only for their suffering would violate Weilian attention as it reduces them to a singular reading.

The unresolvable obligation to the other's call that creates the ethical bind may help develop a better sense of what attentive curating is or could be. The curatorial in attentive curating exists because of a call made by another exposing their vulnerability as well as their call for justice. However, it cannot finish, round out, resolve, or master its core demand. Instead, it always has to return to it, not as a seasoned or experienced expert, but to learn from the other continually. It is in that regard that the

---

[8]　Self-silencing through decreation creates what Andrea Hollingsworth describes as "profoundly creative" moments of "self-renouncing, total identification" [20] (p. 213).

[9]　As Deborah Nelson offers, decreation destroys the "self-delusion" and "fantasy" of secure subjective agency and expressive ability [21] (p. 29).

term practice (as used in "curatorial practice") lends a helpful connotation in its sense of habitualness, repetition, and lack of resolution. Cha's eloquent description of the coexisting demand to "tell" in a format that may resolve into something resembling a story and the unavoidable inability to do so within the ethical bind resonates with the curatorial and thus informs attentive curating—an expression originating from and composed by the other's call that never finishes the story for them [18] (pp. 96–98). To conclude an encounter with suffering and vulnerability would suggest they have finitude and an answer that the curator can reach when attention and decreation conversely inspire the recognition that those experiences can never be read in full.

In the curatorial, attention reveals that the other engaged in the encounter risks losing the opportunity to read themselves within the world. In response, attentive curation offers over communicative opportunities to counteract that constant possibility. Weil intimates that an individual's position provides a one-of-a-kind point of access to the world [4] (p. 88). The pathway forged through the attentive person gains meaning when another person can utilize it to mobilize their own perspectives. Curators are always embedded in some aspect of the specific, be it educational and professional backgrounds, cultural experiences, institutional affiliations, and even participation in creative work. In an attentive framework, a curator's position acts not as the sole guide of their work or, conversely, a complete inhibition to ethical practice. Instead, it can be an entry point for the other's engagement and expression. This entails a radical rethinking of both behaviors and the potential of various resources to carry the readings presented by others. Once one performs self-silencing, ethical action transforming the initial interior work into an exterior gesture can occur.

An unresolved question persists as to whether self-silencing can or should be pursued by everyone tackling questions of oppression, non-belonging, and suffering. Regarding the curatorial, and the proposed mode of attentive curating, it is necessary to ask: is it possible that self-silencing would continue to refuse space and opportunity to voices that are already largely marginalized in the arts and curatorial community? In considering key concerns of existing decolonial practices and those seeking to counter inequalities and discrimination, self-silencing may take away from efforts to create greater equity, recognition, and inclusion for alternative perspectives if curators holding those alternative perspectives are restrained from participating in or shaping the conversation. Is self-silencing still working towards justice in those scenarios?

The answer as to who is called to adopt an ethics of self-silencing as it shapes attentive curating involves a consideration of the subject position of the curator. It is important to note that Weil offers attention to people who are not given it by society at large, individuals who are intruded upon by imbalanced and skewed distributions of power. Weil recognizes and reflects on disparities in power that exist through social and institutional reinforcement, a discernment that can also occur before adopting attentive curating and self-silencing. A curator may have experienced or be experiencing forms of silencing, hostility, or violence that require attention. Their voice could be a part of the call to counteract forces causing subjugation or marginalization, meaning that the articulations of their experience are among those that should be bolstered and not necessarily muted [20] (p. 226).[10] Undeniably, there are individuals in the art world that hold forms of power and access that are not made available or as easy to obtain for others based upon factors such as race, sexuality, nationality, and gender identity. They may be most readily in the position to decide to surrender their power and give space over to another's voice and communication.

However, whether decreative ethics in attentive curating should be adopted is not always a simple categorical distinction. Each call, and thus each occasion creating the circumstances for attentive curating, is original and bears its own conditions that may not disqualify self-silencing based on the curator's subject position. Instead, it may challenge how it is understood and how it is enacted.

---

[10] Weilian ethics attend to precarious subjectivities and alleviate the silencing effects of hardship by embracing decreation for the benefit of someone who may not have the means to make that decision themselves. Hollingsworth explains that "According to Weil, what victims need is a bolstering of their being rather than a stripping of it" [20] (p. 226).

The nature of some projects, contexts, and subject matter that the curator finds themselves working with could make self-silencing detrimental as it removes a vital voice that needs recognition. However, the character of other callings could prove that the same curator does have the power and resources to offer over that can further strengthen the voices that the curatorial event is seeking to serve. This might occur when the subject of the curatorial encounter is completely outside of the curator's experience or when the curator realizes that they cannot speak to or on account of the other's articulation of need. Understanding agency, ability, and positioning in relation to each encounter with the other can help discern the viability for attentive curating and decreative ethics in which the curator's role and its very means of operating are given over as a mode of stewardship. In working with incarcerated individuals, the particularities of their suffering and silencing call on non-incarcerated people and those who have never experienced incarceration who use attentive curating to perform self-silencing.

## 4. Establishing Social Roots through Communicative Agency

Weil's publication *The Need for Roots* outlines concrete realizations of attentive justice as she sets her ethics in the social requirement for rootedness [7] (Chapter 4, Rights and Roots, para. 2).[11] As she claims, vital roots are formed by one's "real, active and natural participation in the life of a community which preserves in living shape certain particular treasures of the past and certain particular expectations for the future" [11] (p. 41). When someone faces the silencing effects of suffering, they experience uprooting because they cannot assert their presence and thus cannot fully engage in or receive affirmation from public life. According to Weil, culture plays an essential role in strengthening a sense of belonging and recognition within a community. Exclusion from culture is both a cause and consequence of social uprooting [11] (p. 65).

The prison–industrial complex presents a Goliathan uprooting entity sanctioned by the state. As mentioned, mass incarceration physically removes individuals from their families and communities while also severing the rights and the engagement that civilian life holds. The dehumanization of inmates further disconnects them from a sense of belonging and recognition. In a self-fulfilling cycle, the prison system continues to benefit from the lack of protection of those incarcerated as it becomes increasingly privatized and profitable. An outstanding example is the use of prison labor to manufacture goods and provide services for extremely low wages, money inmates often have to spend back into prisons and corporations present in prisons. Despite the extensive work experience, many inmates struggle to find employment after their release due to their criminal background, even at the same companies that used their labor while they were in prison [22]. It also exists in widespread voter disenfranchisement of incarcerated individuals who lose access to the primary mechanism of determining their political futures.

As Weil describes, placemaking involves having access to and shared ownership of the terms needed to read one's needs and hopes within a community. Rooting occurs when people feel genuinely acknowledged by others and have active input in shared spaces and conditions. Weil states that "transposition" is a necessary factor in rooting [11] (p. 65). Transposition recognizes that everyone has differing needs regarding how they represent themselves through their own articulations. Attending to others' means of reading builds into social justice for the uprooted by growing their communicative power and generating a space of unrestricted expressive possibility [23] (p. 231). When revealing and connecting to peoples' impersonal desire for good, these encounters foster a non-limiting relationality that values each person's agency in the negotiated communicative sphere [23] (pp. 238–239).[12] Once the

---

[11] Weil wrote *The Need for Roots* in 1943 for the Free French as a guide to rebuilding France after liberation from German occupation during World War II. It was later published in 1949 after her death. *The Need for Roots* responds to the colonial and capitalistic forces and abuses of power utilized to disrupt and ravish communities around the world, creating the uprootedness that makes the type of violence realized by World War II possible [7] (Chapter 4, Rights and Roots, para. 2).

[12] Craig T. Maier suggests that these acts move beyond existing social categories when asking "How can uprooted persons create invitational spaces where they can reflect on their common experiences and develop relationships rooted in hope and care for each other—instead of exclusionary spaces based on ethnicity, nationalism, or political ideology?" [23] (pp. 238–239).

community acknowledges their participation, the uprooted can address what their needs are for the realization of justice.

Weil's notion of rootedness is not far removed from existing curatorial considerations. More and more commonly, curators consider the communities that engage with their work when designing their projects. However, providing a means for another's rootedness goes a step beyond contextual adaptation. It fosters a sense of ownership and involvement for participants while acting as a counter to the apparatuses that prevent others from shaping and determining their situation and their course. Simply utilizing the language, cultural representation, or history of the subject matter and associated people without allowing them to help determine those aspects as they see fit would approach appropriation. It risks suppressing others' voices rather than alleviating the pressures of exclusion. Weilian attention and its effect on rootedness instead call for expanding agency in acts of meaning-making so that others' perspectives can become part of a larger narrative from which they were previously omitted.

## 5. Seeking Attention in Prison-Based Curation

Undeniably, existing exhibitions focused on prisons and those engaging incarcerated individuals grapple with questions of expressive visibility for a community deprived of it and the curators' resulting obligation. Attempts at addressing such concerns represent different alterations in curatorial form and content. *Prison Sentences: The Prison as Site/The Prison as Subject (Prison Sentences)*, organized by Julie Courtney and Todd Gilens and shown from 17 May to 29 October 1995, presents a move to break out of art spaces in order to perform an aesthetic and conceptual archaeology of Eastern State Penitentiary and its connection to the broader United States prison system [24] (pp. 4–5). Rather than bringing their investigations into the gallery, as a traditional show might, Courtney and Gilens invite artists to create site-specific installations within the defunct penitentiary and ask audiences to put themselves inside the structure in order to witness the artworks. While many of the artists incorporate their previous experiences working with inmates, their art acts as the primary demonstrations of realities at play within the penitentiary [25] (p. 50). The decision to use artists as the interpreters of the different types of memory and history reverberating in the building unintentionally emphasizes the absence of voices still affected by incarceration. The relation made within the catalog between artists' social marginalization and that of prison populations creates an unbalanced approximation of experience that circles the absence. However, it fails to fully account for it [25] (p. 22).[13] Despite locating the curatorial event in the prison and using its physicality to expose how people might move through and live within it, the structure of *Prison Sentences* recreates some traditional curatorial paradigms that abstract the realities of those living through incarceration.

In recent years, there have been a large number of prison-focused exhibitions that use an almost reversed process, showing art made by inmates in gallery and museum settings. They take various approaches to enhancing visibility by displaying the inmates' work in formal art spaces. Exhibitions of this ilk include *How Art Changed the Prison*, organized by Jeffery Greene and shown at The Aldrich Contemporary Art Museum from 27 January to 27 May 2019 [26], *On the Inside*, curated by Tatiana von Fürstenberg in collaboration with the organization Black and Pink and shown at Abrons Arts Center from 5 November through 28 December 2016 [27], *The Pencil Is a Key: Drawings by Incarcerated Artists*, curated by Laura Hoptman at the Drawing Center from 11 October 2019 through 5 January 2020 [28], and the Museum of Modern Art's (MoMA) *Marking Time: Art in the Age of Mass Incarceration,* organized by Nicole R. Fleetwood as a counterpart to her book of the same name, curated by Amy Rosenblum-Martín, Jocelyn Miller and Josephine Graf, and on display from 17

---

[13]  Eileen Neff creates the comparison, writing "[The artists] own often marginal status in society suggests an ironic link that, while not accounting for their presence here, reinforces this site's refusal to be read in only one way. Whether working out of society's darkest corners or celebrated at its cultural center, the artist remains a vital figure for our lives" [25] (p. 22).

September 2020 to 4 April 2021 [29].[14] Importantly, such gallery shows demonstrate efforts to reveal critiques and commentaries on the conditions of the United States' criminal justice system and the individuality, hopes, and humanity of those incarcerated. However, as Zachary Small describes in their article "What Curators Don't Get About Prison Art," exhibitions of artworks made by prisoners often do not successfully resist or interrupt the functions of the prison–industrial complex that profit at the expense of people experiencing incarceration [30]. One of the strongest points Small makes is that falsely understanding art spaces to be "neutral" neglects how several of those hosting such prison-focused exhibitions are supported financially or through board service by individuals who profit off of prison privatization [30]. At least to some degree, the art institutions become implicated in the cycles that grow and support the prison–industrial complex while negatively impacting the inmates that curators seek to support through their exhibitions [31,32].[15]

Gallery and museum-based shows, which are not universally accessible to all members of the public (and are physically inaccessible to artists who are still incarcerated), can also remove or misrepresent the context from which the artists experience social silencing and from which they create. The trickiness of exposure emerges, as it can bring their experiences to light while also creating increased vulnerability for participants who face hostility from the public due to their prison or crime affiliation. As Small describes, galleries bear the tensions of identifying artists when presenting participants as inmates of certain institutions to contextualize the artworks, which can make their prison sentence a central and masking characteristic [30]. Alternatively, they can provide some identitary concealment to prevent backlash. However, the choice to limit information about the artists can increase a sense of anonymity and disconnection from their work [30]. Furthermore, Small points out the problematic nature of restrictions or lack of transparency around the ways that inmates are compensated for their artworks. In some cases, they are not compensated at all, and at other times they receive payments to their commissary accounts, which they can use to pay for supplies or phone calls in prison [30]. As Fleetwood discusses, even when their art is used in projects carried out by non-profit organizations raising money for prison-reform initiatives and advocacy, inmates can lose control of their work. She shares the story of an artist she has been working with who "did a lot of art while she was in prison in Ohio, wrongfully convicted for 23 years, and she doesn't know where most of that art is because it became part of a larger circulation of prison art that benefits organizations. That kind of circulation doesn't always benefit the artist who produces the work" [33]. In the case of prison-based curation, the reception of the incarcerated participants' messages and the honest witnessing of causes and realities of their suffering risk being obscured or undermined when the galleries and museums the shows take place in participate in the systems against which the artists are speaking out.

Attentive curating does not have a point-by-point resolution to each of the issues that arise in curation working with incarcerated individuals. However, it does provide a means of rethinking the curatorial in a way that might prove conducive to greater resistance and openness. With its attendant decreative ethics, attentive curating seeks how the curatorial can interrupt the systems that wield the type of force causing vulnerability and suffering so that the other can genuinely be recognized and heard without a corresponding infringement on or assumption of their presence. It requires a critical look at what factors of curatorial positioning might also propagate those types of forces directly or indirectly, masking the other's presence. It reflects on what can be undone to prevent such obscuring from happening. Since the attentive curatorial position surrenders itself to the service of the other and takes form from the nature and needs of their call, it maintains the ability to depart from its familiar and

---

14 *On the Inside* also ran at Craft Contemporary in Los Angeles from 2 June until 8 September 2019.
15 The MoMA has recently faced this very criticism: on 30 October 2019, in advance of the exhibition *Theater of Operations: The Gulf Wars 1991–2011*, artist Phil Collins withdrew a piece meant to be included in the show in support of the ongoing efforts to have MoMA board member Larry Fink divest his company BlackRock from private prisons and detention centers [31]. Fellow artists of the exhibition also penned a letter asking the museum and its trustees to divest from companies profiting from private prisons and military groups [32].

established contexts and methods. This detachment from some of the institutions, spaces, and modes of curating which also offer the largest platforms to elevate voices to a broader audience may be difficult. However, the detachment that comes with the choice to adopt attentive curating promotes a continuous effort to exist in alternative ways while counteracting the systems that foil attempts at witnessing and visibility. It can facilitate the movement out of traditional or formal art spaces that stipulate their own conditions as foundations for curation and are entrenched in certain social and economic structures that affect the subject of the encounter.

The concerns of exposure within prison-based curation are not entirely resolvable in an attentive framework. As stated, attentive curating seeks to create spaces of genuine witnessing that prevent disruptive and harmful intrusions. Despite this, questions as direct as whether or not to list the artists' names at the risk of the public shaming or disparaging the participants do not have an easy answer. Attentive curation, just as much as other forms of curating, cannot control public reception and response. The inability to protect from this potential injury contributes to the incomplete nature of the ethical obligation. However, in receiving the other's perpetual call to be witnessed, attentive curating can balance the risk of exposure by exercising greater placemaking for the other within the curatorial act so that they have the ability to direct and shape their exposure as they see fit. By giving them the means of participation, attentive practices can act as a rooting site that can, hopefully, inspire other rooting sites through exposition to the public. This may entail a bridging of curation with other concrete needs communicated by participants, such as access to the public through expanded media, ways to move around the high costs of prisons, attention to policy and legislation, and many other factors both familiar and unfamiliar to curatorial work. The inclusion of a correspondence system in *On the Inside*, in which viewers can share comments with the artists through text-messages that the organizers later print and mail out to the artists' prisons, provides an example of the possibility of such bridging [34].

## 6. To Send a Kite: A Case Study in Attentive Curating

The prison-focused project Thousand Kites substantiates the potential for attentive ethics in curatorial practice. Started in 1998 by radio DJs Nick Szuberla and Amelia Kirby and run by the non-profit Appalshop through 2011, it utilizes the radio and the internet as transmission and transposition devices to break through a force of social injustice and silencing. The understanding informing the whole project is that marginalized incarcerated voices can establish a greater sense of rootedness through communicative access to the outside world. It allows those overlooked, stereotyped, or even demonized due to their incarceration to challenge and diversify readings of inmates and the criminal justice system both within and beyond prison walls. The name of the project itself underlines such a form of attention as it comes from the prison slang term "to shoot a kite," meaning to send a message [35,36] (p. 108). Although they are the organizers, it is not Szuberla's and Kirby's kite. Rather, it takes flight from inside the community served by the project, revealing a reality that participants can articulate and share while Szuberla and Kirby keep it airborne.

Thousand Kites responds to the construction of Wallens Ridge and Red Onion State Prison, two Supermax facilities built in Virginia as "a quick fix" approach to creating jobs in Appalachia [37]. To fill empty cells, the Virginia Department of Corrections sells inmate transfer contracts to other states around the country [38].[16] Jamie Fellner of the United States Program of Human Rights describes this system of trading as a profit-driven commodification of prisoners [38]. The department has also expanded terms of eligibility for placement to include convictions not traditionally handled in such facilities [38]. The racialization of mass incarceration is one of the many outstanding issues in the Appalachian prisons. Many of the incarcerated individuals relocated to the Supermax prisons in Virginia are Black, and a large portion moved from cities to a largely white, rural area without prior notification [38]. Prisoners and families reveal that the relocations cause a strong sense of displacement,

---

[16] Wallens Ridge alone housed individuals from Connecticut, New Mexico, and even Hawaii [38].

non-belonging, and even conflict and antagonism among inmates and guards [38]. The fact that many families do not have the means to visit their loved ones who have been transferred across state lines only causes greater isolation and uprooting amid the worsening conditions and mistreatment faced by inmates.

Even before being fully aware of the prisons' conditions, Szuberla and Kirby recognize the transferred prison population's need for belonging. By starting the only hip-hop radio station in Appalachia to accommodate the prisoners' musical tastes, they make space for the new community within one of the few shared channels of entertainment available to them [37]. Letters sent to the station from the radio show's growing listener base reveal experiences of racism and human rights violations within the prisons, including the use of excessive force, aggressive and controversial restraint methods, and even deaths said to be caused by the aforementioned tactics and slow responses by prison staff [35–38]. Szuberla emphasizes that they do not write believing the radio show will fix their situation; instead, letters are attempts at "trying to figure out where they were culturally, trying to figure out where they were in relation to their family" [39]. These correspondences resemble the foundational call in a Weilian ethical relationship to recognize another's affliction. They challenge the DJs to attend to the stories and individuals behind them properly. In response, Szuberla and Kirby display behavior akin to Weilian attention by witnessing prisoners' accounts of injustice and dehumanization without imposing a narrative agenda. Before initiating any awareness campaigns, Szuberla uses mail correspondence and his radio show to play chess with the inmates [35]. They connect over a game that relies on the participation of an autonomous, unpredicted other to make counter moves. This simple act of recognition is reminiscent of Weilian attention from a distance as Szuberla orients himself to attend to the inmates' unique realities rather than use them to form a composite identity or an illustrative tool.

The radio DJs first and foremost act as recipients of correspondences from inside the prisons instead of seeking out individuals or information to fit a message proposed for their own research. Kirby describes: "It wasn't as though we had plotted out a story we were going to tell about prisons. A prison came to our community" [37]. In turn, their platform becomes a viable medium through which contact can occur, and a thorough, thoughtful discussion of the complex situation can begin. The project's growth from one form of social engagement, games, to another, storytelling, allows prisoners, families, and community members greater means to read their realities over the radio airwaves and online [37]. Thousand Kites's multiple platforms allow those incarcerated to engage from a distance when sharing and receiving creative content. In many ways, it operates in an open-access manner, such as with the digital submission portal on their website. Szuberla describes how finding the most affordable and accessible technological avenues to capture and broadcast others' stories creates the project's driving structure [39]. The main conditions for participation are simply access to letter-writing materials, a radio, a telephone, the internet, or other cheap media devices. Support from artists, community members, advocates, and the inmates' families helps Szuberla and Kirby record and broadcast content such as poems and rap songs written by inmates and phone calls from family members [35,36]. Their effort expands to broader audiences through a live stream on their website and a play written in collaboration with inmates, prison employees, community members, and families. The play is composed of a scripted storytelling portion conveying the effects of prison culture, a time for cast and audience members to share their experiences and thoughts of the criminal justice system, and a period of reflection to generate ideas for actions moving forward. The script is available online alongside a facilitation guide so that anyone interested in exploring the issues can download it, stage it, and lead a discussion. *Up the Ridge*, their award-winning documentary exploring the events occurring within Wallens Ridge, is also freely available to watch online. Thousand Kites relies on making the content and stories as widely and freely available as possible. Organizers ensure that the collectively authored project may contribute to nationwide perspectives of the issues of incarceration. In doing so, they help those closest to the issues to contribute to and grow others' understandings of the criminal justice system.

One of the key components of communicative access offered through and continuing beyond Thousand Kites is the Calls from Home campaign. DJs and volunteers help manage phone lines at the radio station so that families can call in and broadcast a message via the radio to their incarcerated loved ones whom they are otherwise unable to speak to or visit [37,40]. It provides a useful means of exchange in the face of the staggering costs incurred when making phone calls to prisons. The fact that it takes place on a public radio show means that all listeners, including community members without the experience of imprisonment, can hear personal and non-statistical realities around the polarizing issue of prison reform. It is a direct and basic means of communication within an overwhelming tide of obstacles undermining the significance of one's ability to tell their child, parent, partner, or any other meaningful person in their life "Hello" and "I love you." It allows families the chance to be witnessed by the broader community that often turns away from their hardship. Simultaneously, it broadcasts statements of recognition to prisoners who do not typically have the chance to hear "I see you" or "I am thinking of you" from the outside.

Thousand Kites is exemplary of attentive ethics because the curatorial aspect is mostly subdued or invisible. Szuberla and Kirby's presences exist mainly in write-ups on the project, their regular DJ duties on the air, conference presentations, or Szuberla's YouTube channel, which he uses to post material for broader viewership. From their position, it resembles the impersonal because their social specificity as organizers and how they see these situations do not define the project. The community of participants generate their own narratives without Szuberla and Kirby performing an interpretive digestion or re-presentation. On the other hand, it is profoundly personal for the contributors who share intimate testimonies of both hardship and hope. Szuberla and Kirby show the possibility of behavior evoking decrease within cultural practices by giving over the mechanisms of acknowledgement to participants. They offer up their resources and skills to serve the expressions of others in the hope that it will counter the forces of suffering and silencing that they face. Szuberla and Kirby's main duty is to encounter the stories shared out of individuals' own volition and keep the communicative event moving outwards. They learn from the accounts coming from their communities, and their research is the ongoing reception of insight delivered by those continually affected by the prison system. The radio show and the online content serve as different continuations of this form of long-term, attentive, conversational learning.

Szuberla and Kirby realize the power of intimate glimpses into individual realities within a broad social and political issue. They believe the effect of these expressive sites to be the driving impetus for wider reform. The project creates open spaces of engagement that honor participants' direct connection and unparalleled knowledge of the conditions of prison life without allowing their criminal sentences to solely define the reception of their messages or dismiss their contributions altogether. Thousand Kites manifests its notion of communicative justice through the nature of the curatorial event, which circumvents systems of silencing to broadcast voices outward, instead of simply recommending what justice could be. What started as a basic project of bringing awareness to suffering individuals led to the galvanization of a community and organized efforts for change. Thousand Kites demonstrates how the ethical approach adopted from the earliest stages of the project support the ongoing rooting of participants.

## 7. Conclusions

As evidenced in Thousand Kites, attention can be generative in curatorial service. Non-action is not the same attitude as inaction. Attention can be resistant and catalytic when serving marginalized stories. Curators can listen to hear what those they work with need for community-based rootedness and how to support it through the curatorial mediums at play. Szuberla and Kirby thought through the practical connection they could forge at the earliest stages of Thousand Kites when they incorporated the prisoners' musical taste into their radio programming. The inmates' correspondences then affected the DJs' work on their show. Thousand Kites suggests how attentive ethics can democratize the curatorial process. When shared influence becomes vital and respected, empowerment and agency emerge

and spread horizontally. Weilian attention does not recreate the traditional top-down, authoritative approach to curation, nor does it expect the population served to rely on the goodwill or expertise of the curator. Instead, attentive curation can support others as they articulate their own needs and resistances. The effort behind Thousand Kites demonstrates an approach akin to what Jesson describes as "the struggle to speak of suffering meaningfully while continuing to pay attention honestly" [8] (p. 186). It suggests possibilities for how Weil's ethics may be put into action curatorially as genuine openness and witnessing.

Weil's thinking can help curators continually reflect on the balance between the desire "to speak of suffering meaningfully" and "to pay attention honestly" when working with inmates or engaging with similarly sensitive topics. Those coexisting motivators help to serve stories of affliction and hope. An integral consideration they inspire is how curators can feasibly share creative direction and production so that those they work with may become not just passive subjects but active and central contributors. A project can remain functional while also attending to and respecting individual readings as sources of thematic insight. Balancing creative contribution is both a practical and conceptual task. Relatedly, form and modes of curatorial work can be carefully considered for potential and, perhaps, unintentional points of exclusion that can isolate or hamper others' voices and readings. Another aspect to return to in such work is the impersonal as a form of non-homogenizing listening and presentation. Curators are not solely representing others, nor are they fully identifying with their work's subject matter. They do not have to sustain ideological boxes that stifle shared engagement and dialogue. In the case of prisoners, such impersonality on behalf of the curator makes it possible to attend to inmates' perspectives, which may be challenging or completely distinct from their own, without adopting or resolving them. It also means that the curator does not have to conflate narratives of incarceration to recognize how they contribute meaningfully to discussions of suffering and hope.

A curator can cultivate environments that help generate a sense of rootedness for participants. Such actions include careful research, community interviews, choice of venue, a willingness to make changes throughout the process, even labels and signage that resist the establishment of singular perspectives and authoritative statements. Attention involves all of these but is not reducible to just one. A single dimension of the project does not represent all the stages in which readings occur and the subject is articulated. The ethical attention proposed within this essay acts in several ways as an antithesis to the types of cognitive work individuals are often called to do in the contemporary moment. Instead of concentrating on quick turnaround, multitasking, and rapid production, Weilian attention gives full, undistracted focus to another and patiently waits to receive. This type of attention and related curatorial projects built around it are not profit-driven and cannot be defined by how broadly consumable it may be. Output and tangible results are not easy to measure with traditional gauges of value, such as the number of tickets sold, sales on artworks, or even critical reception. Curators submerged within a magnitude of materials and questions can use earnest reflection to discern whether they can maintain the integrity of attention within their curatorial roles. Szuberla's guiding questions resonate in such an ethical space: "What are the stories in your community that people don't want to talk about? What media tools can you use [to spark that dialogue]?" [37]. His access to a method of broadcasting and his ability to acquire resources and create collaborations proved to be his mechanisms of stewardship. Curators do not work within vacuums as they create with and for others, meaning they can critically think about how their practice can foster responsible and uplifting interactions for those whose plight needs voicing.

**Funding:** This research received no external funding.

**Conflicts of Interest:** The author declares no conflict of interest.

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
