# Peer review of "To Send a Kite: Simone Weil’s Lessons in Ethical Attention for the Curator"

_philosophies, doi:10.3390/philosophies5040032_

Round 1
Reviewer 1 Report
There are a number of dimensions lacking in this contribution. The paper requires further reading and research and major revisions. The paper ignores a long history of curatorial theory grappling with questions of social justice, witnessing, institutional racism and social inequality. Curating and research-driven practices (that include curatorial work) within the expanded field of art have been concerned with these permanent crises and have constructed ways to approach them from ethical, caring and committed viewpoints that nonetheless adopt concrete positions that think past the individual figure of the curator. Curatorial work as a kind of witnessing or reconstruction of evidence (in the work of the research group Forensic Architecture for example) or as a kind of hearing and courtroom situation (Judy Radul’s World Rehearsal Court for example) are just the tip of the iceberg and random examples from a long history of engagement. The momentum of decolonial discourses in contemporary curating is also worth considering as a force that is shaping equity in the field and contributing to the development of curators who are more aware of the injustices of this world.
What is important to note is that by strongly pitting Weil’s ethics of attention against an unidentified curator figure and suggesting the latter take up Weil’s ethics to become an attentive curator you are conjuring a particular caricature of curators prominent in the 90s to late 2000s. This is the stereotype of the blockbuster exhibition curator wandering around the world ‘unrooted’ selecting artists whom they’ve barely met for biennials or similar events. Although this figure might very well still be alive it is safe to say it is no longer the norm, this is primarily because of the vast number of curators graduating from the many curatorial programmes now available and the structural changes that have ensued from this. The economy as a result has changed vastly and widened to include a huge variety of different curatorial positions, scales, attitudes and politics – many of them advocating strong ethical and community-based positions. Thus ‘attentive curating’ should be framed within its wider curatorial context and not only in relation to Weil and the two case studies, the risk of not doing this labour of contextualising is that both the curator figure and the curating you mention will be read as weak metaphors rather than practices you seek to establish in the world. I strongly recommend A) developing strong – almost dictionary-like – definitions of the key ‘new concepts’ you take up in this text, concepts such as ‘attentive curating’ ought be very well defined and not left to the comparative method vis a vis Weil’s ethics. Whenever there is such a concept, I suggest sculpting a definition that builds on Weil but does not solely depend on her thinking. B) contextualising your ideal of ‘attentive curating’ in relation to curatorial-ethical discussions and debates rather than in comparison to an ‘abstract other’ curator. This will demand more work for sure, but the paper will improve significantly.
Also of importance, I highly recommend contextualising the curatorial work with marginalised incarcerated voices that the case studies take up within the larger context of the prison-industrial complex. Not doing so presents a problem to my mind. You are omitting the conditions of possibility of such curatorial work in the first place, the conditions of possibility of such work bring in a complex set of questions that unfortunately or perhaps fortunately challenge the agency of Weil’s ethical framework. It is advisable to confront and try to solve such challenges. Please contextualise, ask what can Weil’s ethics do if the conditions of possibility for working with the incarcerated, underprivileged, criminalised are this: https://www.thenation.com/article/archive/prison-art-shows-essay/ . You risk a certain patronising of the subjects you are trying to give voice to if you don’t, please give more attention to systems, infrastructures, institutions rather than to individuals when further developing this essay.
Finally, I would like to point out that using your words 'self-silenced' selfless almost mute curator you are advocating for might be a very contradictory idea to begin with, for is not withdrawing one's voice as a tactic a form of authorship in the first place? Such contradictions perhaps need to be highlighted and embraced and taken to their conclusions rather than remain unaddressed as they are at the moment.
Reviewer 2 Report
The author makes a significant contribution to Weil studies in this piece. The application of Weil's thought to both curation and incarceration is original and helpful. The explanation of Thousand Kites is beautiful. I take it as my responsibility as a reviewer and expert on Weil to make the Weil section as clear and incisive as possible so that it sets up perfectly the section on Thousand Kites. To that end, three specifications would improve the piece:
(1) It is important to recognize subject position when writing about Weil's ethic of 'self-silencing', as the author puts it. While Weil often sought to silence herself, part of her effort included lifting up other voices, for instance voices of forced laborers from French Indochina (while she was in Marseilles, she attended trials). So I would ask for specification here: Who should adopt an ethic of self-silencing? To what extent are race and gender important to that prescription? I take it part of the author's point is that non-prisoners need to silence themselves to make room for the voices of the incarcerated. That point--or the author's version of it--could be made more clearly.
(2) It would be helpful if the author explained the dates and contexts of Weil's texts. A few sentences on the context of her writings would be very helpful for backing up claims such as that The Need for Roots in fact develops Weil's ethics of attention. The author cites Gravity and Grace to present Weil's attention. It would be helpful to add when and how those notebook entries were written. (A good example of someone who does this with Weil's writings is Lyndsey Stonebridge in her chapter on Weil in Placeless People.) These contextual notes would allow the author to tie together points of the article. Let me try to explain my point. I understand the author to be critical of how incarceration (enforced by the state) uproots some individuals. Weil was also critical of the state (especially in her early writings, see Oppression and Liberty), and even in The Need for Roots (at times). Noting the politics of Weil's writing would allow the author to convey to the reader that Weil was concerned with the state's role in silencing the powerless, a central concern of this article.
(3) Could the author please clarify their presentation of Levinas and Weil? They write that Levinas's 'view of God's influence... stands apart from Weil's' (5). I am not so sure. Does the author see Weil's development of attention as occurring apart from her spiritual journal and theological contributions? I understand that this journal seeks a quick turn-around time. Still, the author might look at Yoon Sook Cha's excellent Decreation and the Ethical Bind for an extended discussion of decreation and attention in Weil. In my view, given the audience of the article's intervention (presumably curators and also readers of Weil looking to see how her concepts can be applied), it would be worth spending more time slowly describing how an ethical bind (of self and others) forms in Weil's thinking, which Cha discusses at length. I would find that more helpful than working through the Levinasian criticism and the Kantian distinction (footnote three). The authors own voice can--and should--come through more here.
Because reviewers often present themselves as paragons of clarity (yikes), I will try to reiterate my suggestions as clearly as possible: (1) answer the question To whom is Weil prescribing an ethics of self-silencing? (2) Track the context out of which the writings of Weil you cite emerged. (3) Tone down the secondary literature in parts 2-4; keep more of the Dietz, Roberts, Nelson, and Gayman in the notes to let the non-specialist or non-academic move through the piece more easily. To accomplish (1-3), engaging with (though keeping this research in the background of the essay) Cha's Decreation and the Ethical Bind and Stonebridge's chapter on Weil in Placeless People would be helpful.
Round 2
Reviewer 1 Report
Thank you for your additions, revisions and reformulations. The paper has vastly improved and I can feel very comfortable accepting it for publication. The contextualisation and addition of new theoretical references as well as new examples/case studies has transformed this paper and added the necessary contextual information and critical insights for your theory to hold up. I have no further comments of any rigorous demand but in terms of fine detail I think lines 78 - 83 would benefit from further clarification as to who is the 'creature / creator' when grafted onto the curatorial setting you are describing and line 581 - 585 could also do with some rewording for clarity since something is not immediately coming across. Otherwise the paper sound and communicates its position clearly.
